# A facile approach for the in vitro assembly of multimeric membrane transport proteins

Erika A Riederer[1†], Paul J Focke[1†], Elka R Georgieva[2,3‡], Nurunisa Akyuz[4§], Kimberly Matulef[1], Peter P Borbat[2,3], Jack H Freed[2,3], Scott C Blanchard[4], Olga Boudker[4,5], Francis I Valiyaveetil[1*]

[1]Department of Physiology and Pharmacology, Oregon Health & Science University, Portland, United States; [2]Department of Chemistry and Chemical Biology, Cornell University, Ithaca, Unites States; [3]National Biomedical Center for Advanced Electron Spin Resonance Technology, Cornell University, Ithaca, United States; [4]Weill Cornell Medicine, New York, United States; [5]Howard Hughes Medical Institute, Maryland, United States

*For correspondence:
Francis Valiyaveetil;
email: valiyave@ohsu.edu

[†]These authors contributed equally to this work

Present address: [‡]Weill Cornell Medicine, New York, United States; [§]Department of Neurobiology, Harvard Medical School, Boston, United States

**Abstract** Membrane proteins such as ion channels and transporters are frequently homomeric. The homomeric nature raises important questions regarding coupling between subunits and complicates the application of techniques such as FRET or DEER spectroscopy. These challenges can be overcome if the subunits of a homomeric protein can be independently modified for functional or spectroscopic studies. Here, we describe a general approach for in vitro assembly that can be used for the generation of heteromeric variants of homomeric membrane proteins. We establish the approach using $Glt_{Ph}$, a glutamate transporter homolog that is trimeric in the native state. We use heteromeric $Glt_{Ph}$ transporters to directly demonstrate the lack of coupling in substrate binding and demonstrate how heteromeric transporters considerably simplify the application of DEER spectroscopy. Further, we demonstrate the general applicability of this approach by carrying out the in vitro assembly of VcINDY, a $Na^+$-coupled succinate transporter and CLC-ec1, a $Cl^-/H^+$ antiporter.
DOI: https://doi.org/10.7554/eLife.36478.001

## Introduction

Movement of ions and small molecules across cellular membranes takes place through transport proteins such as ion channels and transporters. A frequently observed feature of transport proteins is that they form multimeric assemblies of identical subunits in their native state (*Veenhoff et al., 2002*; *Forrest, 2015*). There are challenges in spectroscopic investigations of homo-multimeric (or homomeric) proteins that arise due to the presence of multiple identical subunits. One such example is in using Double Electron-Electron Resonance (DEER) or Fluorescence Resonance Energy Transfer (FRET) spectroscopy with homomeric proteins. These spectroscopic techniques, which are commonly used for detecting structural changes accompanying function, rely on site-specific labeling with appropriate spectroscopic probes, typically at introduced cysteine residues. In homomeric proteins, such efforts result in the labeling of each of the individual subunits and therefore the presence of multiple probes in the protein, which can complicate the spectroscopic measurements. It is also not feasible to label a single subunit in a homomeric protein, which hinders the use of these spectroscopic approaches for investigating the structural changes that take place within individual subunits. The presence of multiple identical subunits in transport proteins also raises the pertinent question of whether there is a functional coupling or 'crosstalk' between the subunits. These challenges in

investigating homomeric proteins can be overcome if we generate heteromeric variants in which a single subunit (or a set of subunits) can be selectively modified. Labeling a single subunit with spectroscopic probes will facilitate the investigations of the intra-subunit structural changes, while functional perturbation of a subset of subunits in the assembly will enable investigations of the functional coupling between protomers. Thus, the ability to generate a heteromeric variant of a homomeric protein will be a great asset.

Cellular expression of heteromeric proteins containing wild type and mutant subunits has been carried out using either concatenation or co-expression approaches. In the concatenation approach, protein expression is carried out using a DNA construct in which the genes for the wild type and mutant subunits are concatenated into a single reading frame. Expression of this concatenated construct yields a heteromeric membrane protein in which the subunits are linked together. This approach has been widely used for the expression of heteromeric ion channels in *Xenopus* oocytes for functional studies (*Gordon and Zagotta, 1995*; *Yang et al., 1997*). There are only a few examples where multimeric membrane proteins encoded in a single polypeptide have been purified for functional and spectroscopic studies (*Raghuraman et al., 2012*; *Lim et al., 2016*; *Last et al., 2016*; *Wang et al., 2016*). In the co-expression approach, the wild type and the mutant subunits are co-expressed in the same cell (*Becker et al., 2014*; *Lu et al., 2017*), wherein the random mixing of the subunits during assembly results in the generation of heteromeric and homomeric proteins. Placing different purification tags on the wild type and the mutant subunits allows the purification of the desired heteromeric protein. Both of these approaches frequently suffer from very low yields.

The alternate approach is to assemble the heteromeric proteins in vitro by using a mixture of wild type and mutant subunits. In this manner, a mixture of heteromeric and homomeric proteins is produced from which the desired heteromeric complex can be purified. The advantage of the in vitro approach is that the ratio of the wild type and the mutant subunits can be adjusted to ensure formation of the desired heteromeric protein in good yields. The heteromeric assembly can be carried out through a coupled folding and oligomerization process starting with fully unfolded subunits. However, refolding of membrane proteins is very challenging, and there are only few reports in the literature (*Neumann et al., 2014*; *Popot, 2014*). In lieu of complete refolding, we envisioned an approach in which we dissociate the native wild type and mutant multimeric proteins into subunits and then use a mixture of the dissociated subunits for assembly of the hetero-oligomers. We anticipated that this dissociation/reassociation approach should be generally applicable and also provide higher yields compared to complete refolding.

Here, we describe a simple methodology for the in vitro reassembly of functional multimeric membrane proteins from dissociated subunits. We develop the methodology using the archaeal glutamate transporter homolog $Glt_{Ph}$. We demonstrate the utility of the approach by assembling heteromeric $Glt_{Ph}$ transporters that we use to investigate the cross-talk between the substrate binding sites and to evaluate the intra-subunit structural changes in $Glt_{Ph}$ using DEER spectroscopy. Further, we establish the general applicability of the methodology by carrying out the in vitro reassembly of the archaeal transporter $Glt_{Sm}$ and the bacterial transporters VcINDY and CLC-ec1.

## Results

### In vitro reassembly of $Glt_{Ph}$

To develop a protocol for the in vitro reassembly of multimeric membrane proteins, we initially focused on $Glt_{Ph}$, a homo-trimeric sodium coupled aspartate transporter (*Figure 1A*)(*Yernool et al., 2004*; *Boudker et al., 2007*). Each subunit of $Glt_{Ph}$ has a complex topology with eight transmembrane and two reentrant hairpin segments. Remarkably, we have previously successfully refolded $Glt_{Ph}$ from a completely unfolded state by using lipid vesicles to obtain an active native-like protein (*Focke et al., 2015*). Here, we investigated whether $Glt_{Ph}$ can be reassembled in vitro from dissociated subunits using lipid vesicles (*Figure 1B*).

$Glt_{Ph}$ migrates as a monomer on SDS-PAGE, suggesting that SDS dissociates $Glt_{Ph}$ into subunits (*Yernool et al., 2004*). Glutaraldehyde cross-linking confirmed complete dissociation of the trimeric species by SDS (*Focke et al., 2015*). While crosslinking of the native $Glt_{Ph}$ gives a protein band on SDS-PAGE corresponding to the trimer, crosslinking following SDS treatment only yields monomers. For reassembly, we diluted the protein after SDS treatment into buffer supplemented with lipid

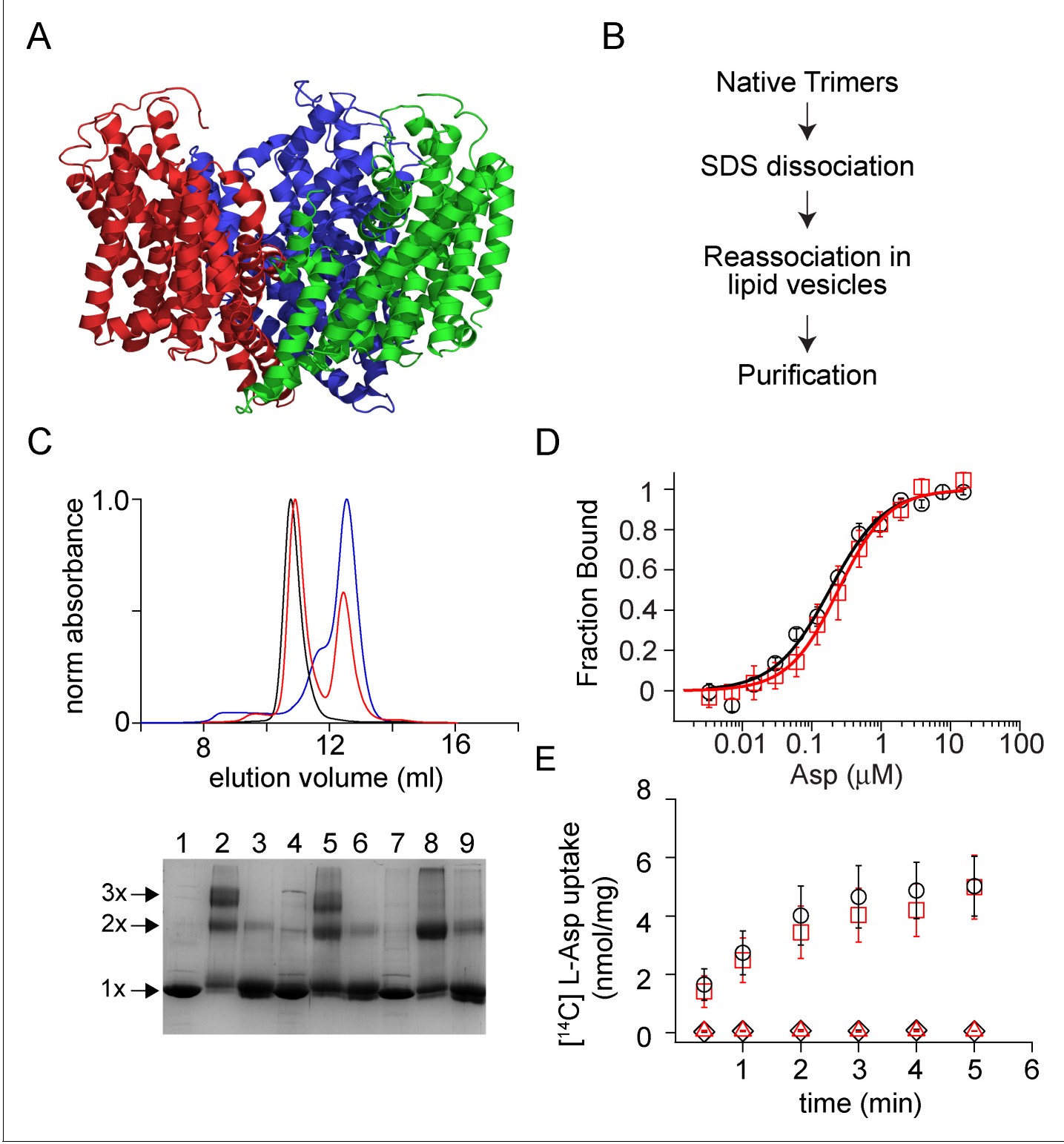

**Figure 1.** Dissociation and reassociation of Glt$_{Ph}$. (**A**) Structure of the Glt$_{Ph}$ trimer (pdb: 2nwx) is shown in ribbon representation. (**B**) Flow chart outlining the strategy used for the dissociation and reassociation of the Glt$_{Ph}$ trimer. (**C**) Size exclusion chromatography of the native Glt$_{Ph}$ (black) and the SDS dissociated Glt$_{Ph}$ following dilution into lipid vesicles (red) and into DDM (blue). Reassembly of Glt$_{Ph}$ takes place in lipid vesicles. Inset, SDS-PAGE gel showing the glutaraldehyde crosslinking of the native Glt$_{Ph}$ (lanes 1–3), reassociated Glt$_{Ph}$ (r-Glt$_{Ph}$, lanes 4–6) and the SDS dissociated Glt$_{Ph}$ subunits (lanes 7–9). Lanes 1, 4, and 7, without glutaraldehyde crosslinking, lanes 2, 5, and 8, with glutaraldehyde crosslinking and lanes 3, 6, and 9, with glutaraldehyde crosslinking in the presence of 1% SDS. The oligomeric nature of the protein band (1x, 2x, and 3x) is indicated. (**D**) Asp binding by

*Figure 1 continued on next page*

*Figure 1 continued*

native and r-Glt$_{Ph}$. Asp binding was monitored by changes in fluorescence of the native (black circles) or r-Glt$_{Ph}$ (red squares) with the L130W substitution. The fraction of the protein bound (F$_{bound}$) is determined by dividing the change in fluorescence upon addition of Asp by the total change at the end of the titration. Solid lines are fits to the data using the equation described in Materials and methods with K$_D$ values of 183 ± 16 nM for native Glt$_{Ph}$ and 216 ± 15 nM for r-Glt$_{Ph}$. The binding assays were conducted in 10 mM Na$^+$. (E) Asp uptake assay. The time course of [$^{14}$C]-Asp uptake for native Glt$_{Ph}$ (black circles) and r-Glt$_{Ph}$ (red squares) in the presence of a Na$^+$ gradient. No uptake is observed for native Glt$_{Ph}$ (black diamonds) and r-Glt$_{Ph}$ (red triangles) in the absence of a Na$^+$ gradient. For panels D and E, error bars indicate standard error of mean (SEM) for n ≥ 3.

DOI: https://doi.org/10.7554/eLife.36478.002

vesicles achieving SDS concentrations below the critical micellar concentration (*Focke et al., 2015*). Consequent glutaraldehyde crosslinking showed the presence of a trimeric band indicating subunit reassociation. We solubilized the reassembled Glt$_{Ph}$ (r-Glt$_{Ph}$) from the lipid vesicles using dodecyl-β-D-maltopyranoside (DDM) detergent and purified it by affinity and size exclusion chromatography (SEC). The SEC elution profile for r-Glt$_{Ph}$ was similar to that of native Glt$_{Ph}$ and crosslinking confirmed the trimeric state of r-Glt$_{Ph}$ (*Figure 1C*). Lipid vesicles were required for the reassociation process, and dilution of the SDS-dissociated Glt$_{Ph}$ subunits directly into DDM did not yield a trimeric species, as indicted by both SEC and cross-linking (*Figure 1C*).

To establish the functionality of r-Glt$_{Ph}$, we examined its ability to bind and transport substrate L-aspartate (Asp). Substrate binding was assayed by monitoring changes in the intrinsic fluorescence of the L130W mutant of Glt$_{Ph}$ upon titration with Asp (*Boudker et al., 2007*; *Hänelt et al., 2015*). The Asp dissociation constant (K$_D$) of the reassembled L130W mutant of Glt$_{Ph}$ was similar to that of the transporter prior to the dissociation/reassociation procedure (*Figure 1D*). We reconstituted r-Glt$_{Ph}$ into lipid vesicles to measure transport activity and observed uptake of [$^{14}$C]Asp into proteoliposomes in the presence of an inwardly directed Na$^+$ gradient (*Figure 1E*). The specific transport activity observed for r-Glt$_{Ph}$ was similar to that of the native transporter.

The key conformational transition of Glt$_{Ph}$ that underlies substrate translocation across the membrane is an elevator-like movement of the peripherally located transport domain within each protomer relative to the central trimerization domain (*Reyes et al., 2009*). Previously, single-molecule FRET (smFRET) was used to visualize these movements in real time (*Akyuz et al., 2013*; *Erkens et al., 2013*; *Akyuz et al., 2015*). These experiments further suggested that the transition frequency determined the overall rate of substrate transport. We used smFRET to compare the dynamics of the r-Glt$_{Ph}$ to the native protein. For these experiments, N378C mutant of Glt$_{Ph}$ was derivatized with Cy3 and Cy5 fluorophores and with a PEG-biotin linker either before or after the reassembly procedure. As previously described (*Akyuz et al., 2013*), the labeled proteins were immobilized on passivated streptavidin-coated microscope slides and imaged using total internal reflection fluorescence (TIRF) to obtain smFRET recordings (*Figure 2A*). In these experiments, movements of two transport domains within the trimers relative to each other are detected. We found that r-Glt$_{Ph}$ in the presence of 200 mM Na$^+$ and 100 µM Asp, sampled three FRET efficiency states, low-, intermediate- and high-FRET centered at ~0.35, ~0.55 and ~0.8, respectively (*Figure 2B*). Similar FRET efficiency states were observed for the native protein, where the low-FRET state was attributed largely to the outward facing state of the transporter. Excursions into the higher-FRET states reflect sampling of the inward-facing conformations (*Akyuz et al., 2013*). The populations of the low-, intermediate-, and high-FRET states for the r-Glt$_{Ph}$ were 76%, 17% and 7%, respectively, in good agreement with the values observed for the native transporter (71%, 16% and 13%, respectively). Transition density plots showed that the r-Glt$_{Ph}$ exhibit conformational transitions at a rate of ~0.03/s, also in good agreement with the transition frequency of the native transporter (*Figure 2C*). In line with these findings, the FRET-state lifetimes were similar in the native and reassembled transporters (*Figure 2D*). These smFRET measurements suggest that the reassembled transporter samples the outward- and inward-facing state with probabilities and frequency similar to the native protein. Collectively, these results show that r-Glt$_{Ph}$ is structurally, functionally and dynamically similar to the native transporter.

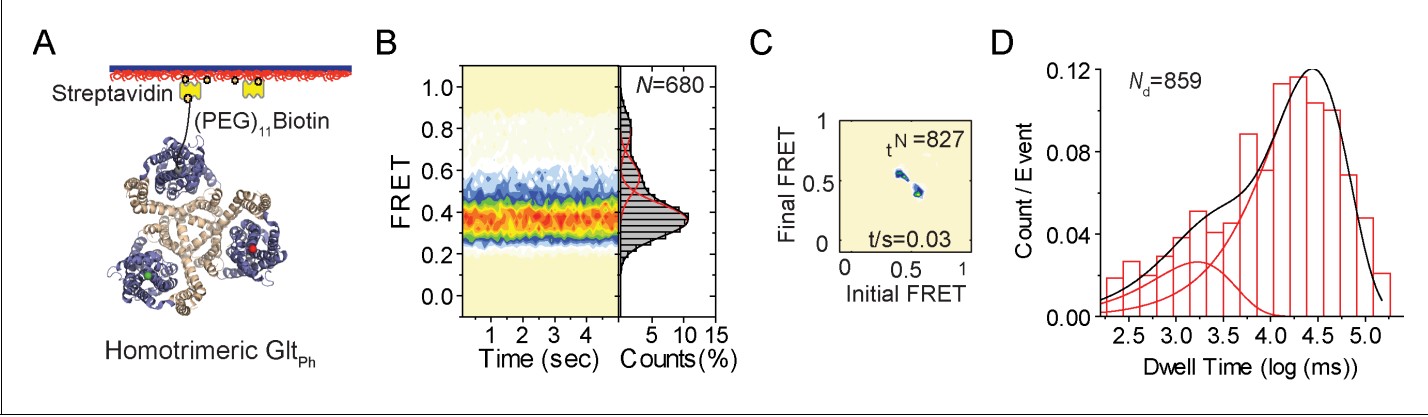

**Figure 2.** smFRET experiments on reassociated Glt_Ph molecules. (**A**) Labeling and surface-immobilization strategy for smFRET experiments. (**B**) Population distributions FRET efficiency population contour plots (left) and cumulative population histograms (right) are shown for r-Glt_Ph. The contour plots are color-coded from tan (lowest) to red (highest population) with the color scale from 0–12%. The population histograms display the time-averaged values and standard deviations. The solid black lines are fits to the sums of individual Gaussian functions (red lines). The number of molecules analyzed (N) is shown. (**C**) Transition density plots show that transitions occur at a frequency of ~0.03/ s. The number of transitions in the dataset ($N_t$) is shown. (**D**) Dwell time distribution for the low-FRET state obtained for r-Glt_Ph shows biphasic behavior and was fitted to a probability density function. The number of dwells in the analysis ($N_d$) is shown.

DOI: https://doi.org/10.7554/eLife.36478.003

## Using heteromeric transporters to evaluate cooperativity in Asp binding to Glt_Ph

A key question in multimeric transporters is whether the protomers function entirely independently or are coupled or coordinated in some manner. In glutamate transporters, several lines of evidence suggest that subunits in the trimer are independent of each other (*Koch and Larsson, 2005*; *Grewer et al., 2005*; *Erkens et al., 2013*; *Ruan et al., 2017*). However, addressing directly whether one subunit senses substrate binding to its neighbor has not been possible. Our methodology allows assembly of heteromeric Glt_Ph transporters in which the ability of individual subunits to bind substrate can be independently manipulated. Thus, we assembled Glt_Ph heteromers containing one reporter and two test subunits (*Figure 3A*). The reporter subunit carried the L130W substitution, which provides a spectroscopic probe for Asp binding (*Figure 3—figure supplement 1*). The test subunits were either wild type or carried the R397A mutation, which lowers Asp affinity by approximately a thousand fold (*Figure 3—figure supplement 2*)(*Verdon et al., 2014*). A comparison of Asp binding to the reporter subunit of Glt_Ph heteromers containing either wild type or mutant test subunits should reveal whether the reporter subunit senses substrate binding to its neighboring subunits.

We reassembled heteromeric Glt_Ph transporters using a 10:1 mixture of the test to the reporter subunits. At this ratio, a random mixing of the subunits ensures that the majority of the transporters assembled will contain either no reporter subunits (73%) or only one reporter subunit (24%), while the fraction with two or three reporter subunits will be negligibly small (3% and, 0.1% respectively). The reporter subunits carried a His_8 tag while the test subunits carried no purification tags. The presence of the His_8 tag on the reporter subunit allowed us to purify only the heteromers containing a reporter subunit (*Figure 3—figure supplement 3*). We measured Asp-binding affinity to the reporter subunit of heteromeric Glt_Ph transporters in the context of either the wild type or the R397A test subunits (*Figure 3B,C*). The binding isotherms were obtained in the presence of 10 mM Na$^+$ ions and fitted to the Hill equation. The $K_D$ values for Asp were 455 ± 61 nM and 356 ± 36 nM for the heteromers with wild type and R397A test subunits, respectively. The Hill coefficients were also similar. Notably, the expected affinity of the R397A mutant in the presence of 10 mM Na$^+$ ions is approximately 238 μM, suggesting that the R397A mutant test subunits remain largely unbound to Asp throughout the titration. This result indicates that Asp binding to the test subunits does not affect Asp binding to the reporter subunit. Therefore, for the first time, we directly demonstrate the lack of coupling in substrate binding to the trimeric transporter Glt_Ph.

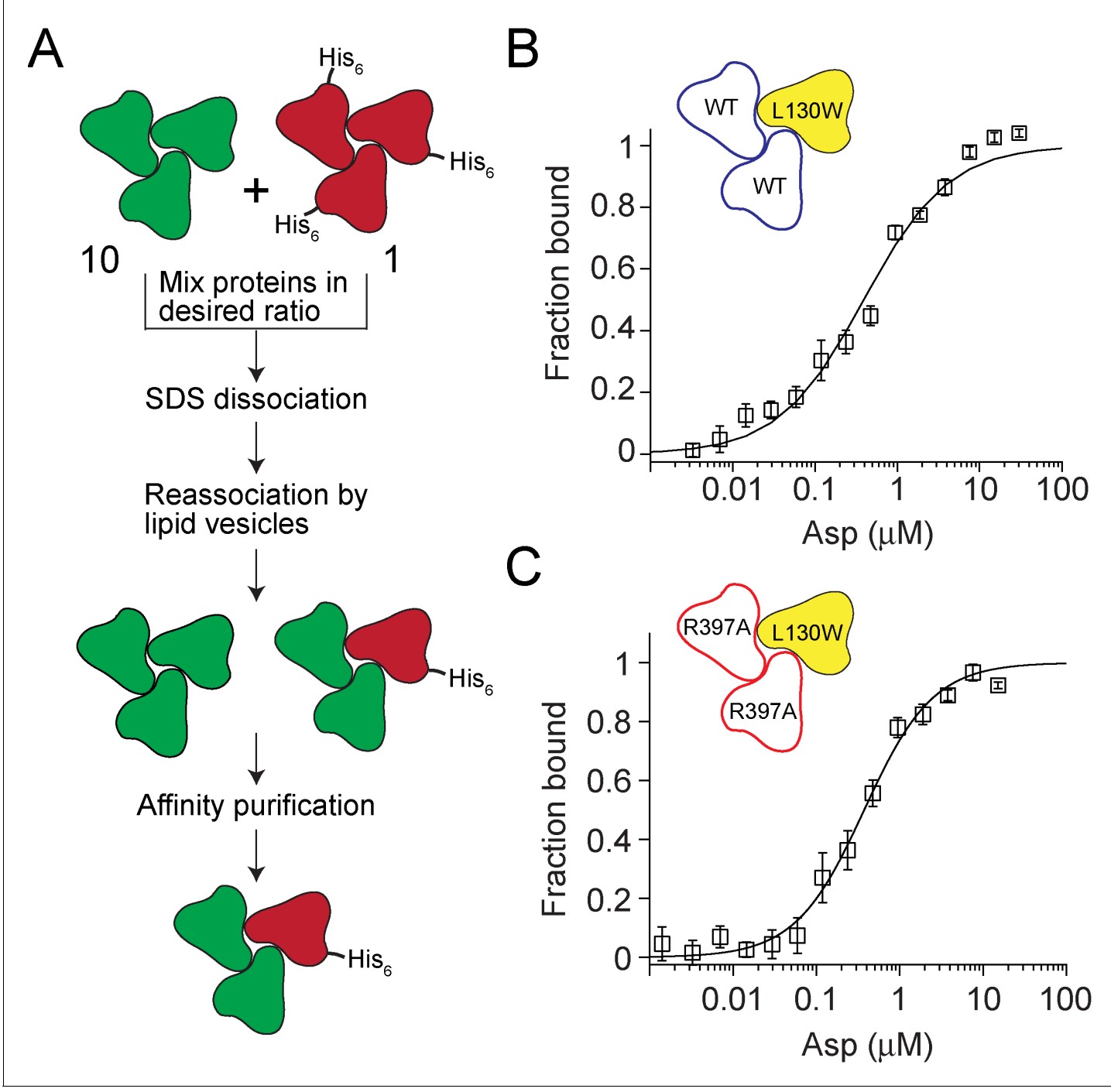

**Figure 3.** Testing crosstalk between Glt$_{Ph}$ subunits in Asp binding. (**A**) Flowchart outlining the strategy used for assembling heterotrimers of Glt$_{Ph}$. The test subunits are depicted in green and the reporter subunits are in red. A 10: 1 ratio of test to reporter subunits is used to ensure that the probability of assembling heterotrimers with more than one reporter subunit is very low. The presence of a polyhistidine tag on the reporter subunit enables purification of the heterotrimers that contain a reporter subuit. (**B and C**) Asp binding to heterotrimeric Glt$_{Ph}$ with test subunits containing the wild-type Asp-binding site (blue, **B**) or with a R397A substitution in the Asp binding site (red, **C**). The reporter subunit (yellow) contains the L130W substitution for monitoring Asp binding to the subunit. Asp-binding assays as described in *Figure 1D*. The solid lines indicate fits to the data using a Hill equation with $K_D = 455 \pm 61$ nM, Hill coefficient = $0.84 \pm 0.08$ for the heterotrimer with the wild-type test subunits and with $K_D = 356 \pm 36$ nM, Hill coefficient = $1.05 \pm 0.07$ for the heterotrimer with test subunits carrying the R397A substitution. Error bars indicate SEM for $n \geq 3$.

DOI: https://doi.org/10.7554/eLife.36478.004

The following figure supplements are available for figure 3:

*Figure 3 continued on next page*

**Figure supplement 1.** Spectral properties of wild type and L130W Glt$_{Ph}$.

**Figure supplement 3.** Assembly of heteromeric Glt$_{Ph}$.

## Using heteromeric transporters to evaluate intra-subunit structural changes using DEER spectroscopy

Understanding the transport mechanism requires determination of the structural changes that take place during function. In recent years, pulse dipolar ESR spectroscopy (PDS), in particular the double electron-electron resonance (DEER) method, utilizing spin labeling has become the method of choice for monitoring structural transitions in biological macromolecules and complexes by reporting on distances in the range from about 10 Å (*Fafarman et al., 2007*; *Borbat et al., 2014*) to as large as 80 Å (*Borbat and Freed, 2007*; *Georgieva et al., 2008*; *Jeschke, 2012*; *Borbat et al., 2014*) or even longer (*Schmidt et al., 2016*) in some cases. It has been widely applied to follow conformational changes in membrane transporters. (*Borbat et al., 2007*; *Hänelt et al., 2013*; *Georgieva et al., 2013*; *Dastvan et al., 2016*; *Khantwal et al., 2016*) In these experiments, the transporters are site-specifically spin-labeled at introduced cysteine residues and dipolar coupling between pairs of spin-label unpaired electron spins is measured. The dipolar coupling depends on the inter-spin distance r, as $1/r^3$, making the method quite an accurate tool for accessing minute distance variations.

The use of DEER spectroscopy for distance measurements in transporters comprised of more than two subunits is problematic, however, as these proteins present multi-spin systems if spin-labeled using standard approaches (*Endeward et al., 2009*; *Dalmas et al., 2012*; *Georgieva et al., 2013*; *Hänelt et al., 2013*). It is especially challenging to use DEER to probe the structural changes taking place within a single subunit, because dipolar coupling is present not only between the spin labels within this subunit, but also between those in different subunits. In Glt$_{Ph}$, the key transition involves the movement of the transport domain relative to the scaffold domain by as much as 15 Å. Probing this movement requires spin labels in both these domains. However, the introduction of Cys mutations into each domain and subsequent labeling of Glt$_{Ph}$ yields six spin labels per the trimeric protein (*Figure 4A*). In the simplest case, with all subunits being in the outward-facing state, we expect up to five unique distances between spin labels, depending on symmetry. Consequently, the distance distributions, generated from DEER signals, are very broad and difficult to extract, being further complicated by non-linear effects (*Jeschke et al., 2009*) and less than ideal spin-labelling efficiency. The use of heteromeric Glt$_{Ph}$ transporters, in which only a single subunit is labeled (*Figure 4A*), would be a great simplification directly yielding, for example, the distance distributions between the probes in the scaffold and transport domains of a single protomer.

To demonstrate the utility of heteromeric Glt$_{Ph}$ for DEER spectroscopy, we assembled a Glt$_{Ph}$ trimer from a 10:1 mixture of wild-type (i.e. cysteine-less) subunits and V216C/I294C mutant subunits bearing cysteine mutations in the scaffold and transport domains. The heterotrimers were purified using the His$_8$-tag on the V216C/I294C subunit and labeled with the MTSL nitroxide [(1-oxyl-2,2,5,5-tetramethylpyrrolidin-3-yl) methyl methanethiosulfonate] spin label. As a control, we also prepared a homomeric V216C/I294C spin-labeled trimer Glt$_{Ph}$. In the presence of Na$^+$ ions and Asp, homomeric V216C/I294C mutant yielded, as expected, broad weakly structured distance distributions, spanning the distance range from ca. 40 to 70 Å (*Figure 4B*). In contrast, the heteromeric protein exhibited a single narrow peak at 52 Å. Remarkably, these measurements suggest that under our experimental conditions the V216C/I294C mutant subunit predominantly populates the inward facing state. In this conformational state, the C$_\alpha$-C$_\alpha$ distance between residues 216 and 294 is 48 Å based on the crystallographic model, in good agreement with the DEER measurement. The corresponding distance in the outward facing state is 34 Å. In conclusion, using a heteromeric trimer greatly simplified the experimentally obtained distance distributions and allowed for straightforward determination of the conformational state of the labeled protomer.

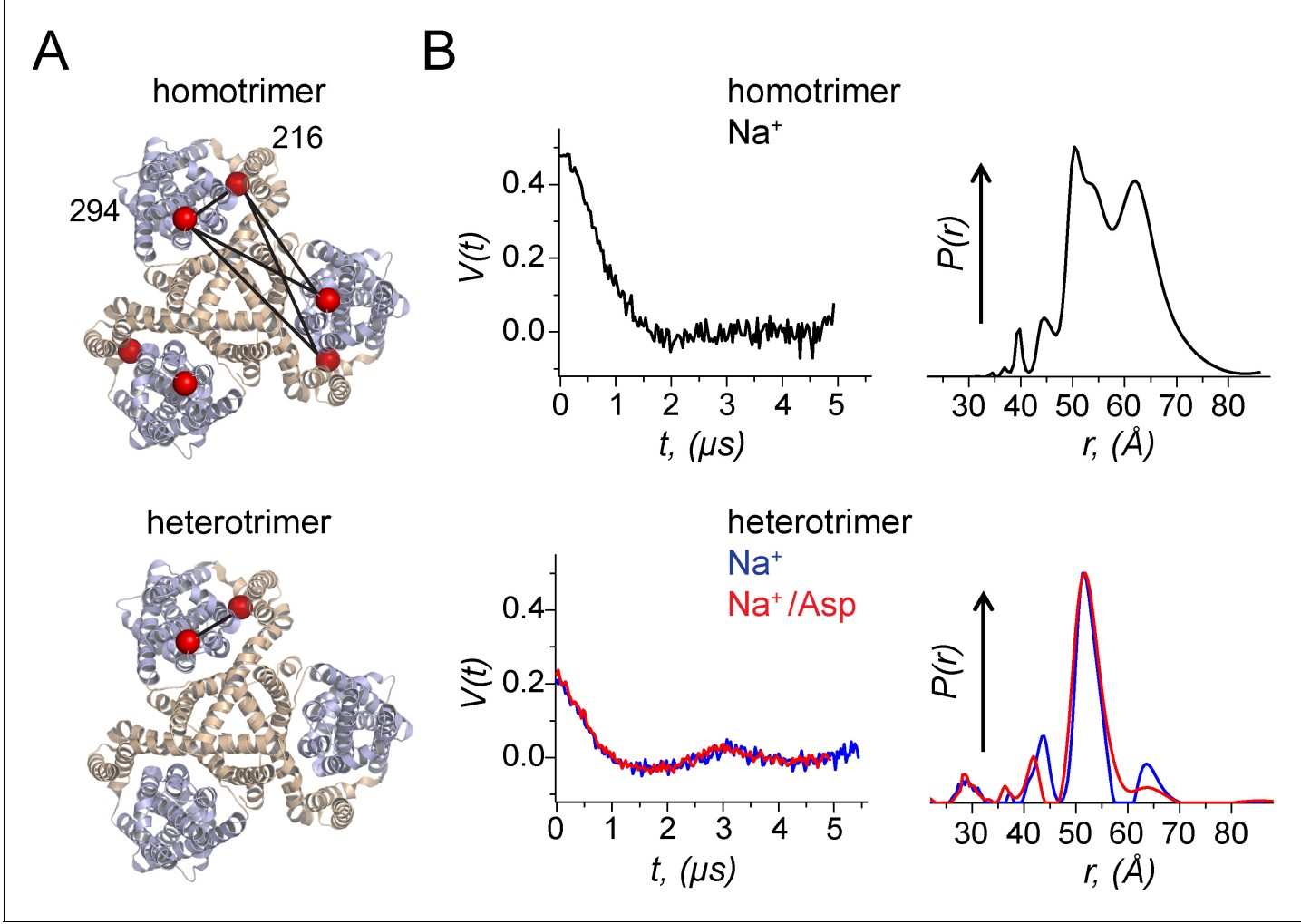

**Figure 4.** Using a heterotrimeric Glt$_{Ph}$ to probe movements of the transport domain using DEER. **(A)** A homomeric and a heteromeric Glt$_{Ph}$ transporter are shown in ribbon representation. The scaffold domain is colored in wheat while the transport domain is colored in light blue. The C$_\beta$ atoms of the Cys residues at 216 and 294 labelled with spin probes are shown as red spheres. All subunits in the homomeric Glt$_{Ph}$ carry the Cys substitutions while only one subunit in the heteromeric Glt$_{Ph}$ carries the Cys substitutions. The distances monitored in the DEER experiment are indicated by solid lines. **(B)** Background-corrected DEER amplitude, $V(t)$ vs. evolution time $t$ (left). The data are shown for: Glt$_{Ph}$ homotrimers with spin labels at both positions in each of the protomers prepared with 200 mM NaCl (black); Glt$_{Ph}$ heterotrimer with spin-labels just in one of the protomers prepared with either 200 mM NaCl (blue) or 200 mM NaCl/300 µM aspartate (red). The data are plotted to have the 'DEER modulation depth' at $t$ = 0 and decay to zero value asymptotically (see Materials and methods). Larger modulation depth, that is $V(0)$ indicates larger number of coupled spins. Inter-spin distance distributions, $P(r)$, reconstructed from the above DEER data, are plotted in respective colors (right). All $P(r)$'s were normalized to common value at the maxima. Glt$_{Ph}$ heterotrimers show much simpler $P(r)$ as compared to convoluted result for homotrimers.

DOI: https://doi.org/10.7554/eLife.36478.008

## General applicability of in vitro reassembly

To investigate whether our in vitro subunit reassembly procedure is applicable to other multimeric transporters, we initially tested the procedure on Glt$_{Sm}$, a glutamate transporter homolog from *Staphylothermus marinus* that shares 58% sequence identity to Glt$_{Ph}$ (*Figure 5A*)(*Jaehme and Michel, 2013*). We observed that SDS dissociated the Glt$_{Sm}$ transporter and that the dissociated subunits can be reassembled back to the trimeric state upon dilution of SDS coupled with protein incorporation into lipid vesicles (*Figure 5B*). The reassembled Glt$_{Sm}$ transporter was functionally similar to the native protein, suggesting that the disassembly/reassembly procedure was well tolerated by Glt$_{Sm}$, similar to Glt$_{Ph}$ (*Figure 5C*).

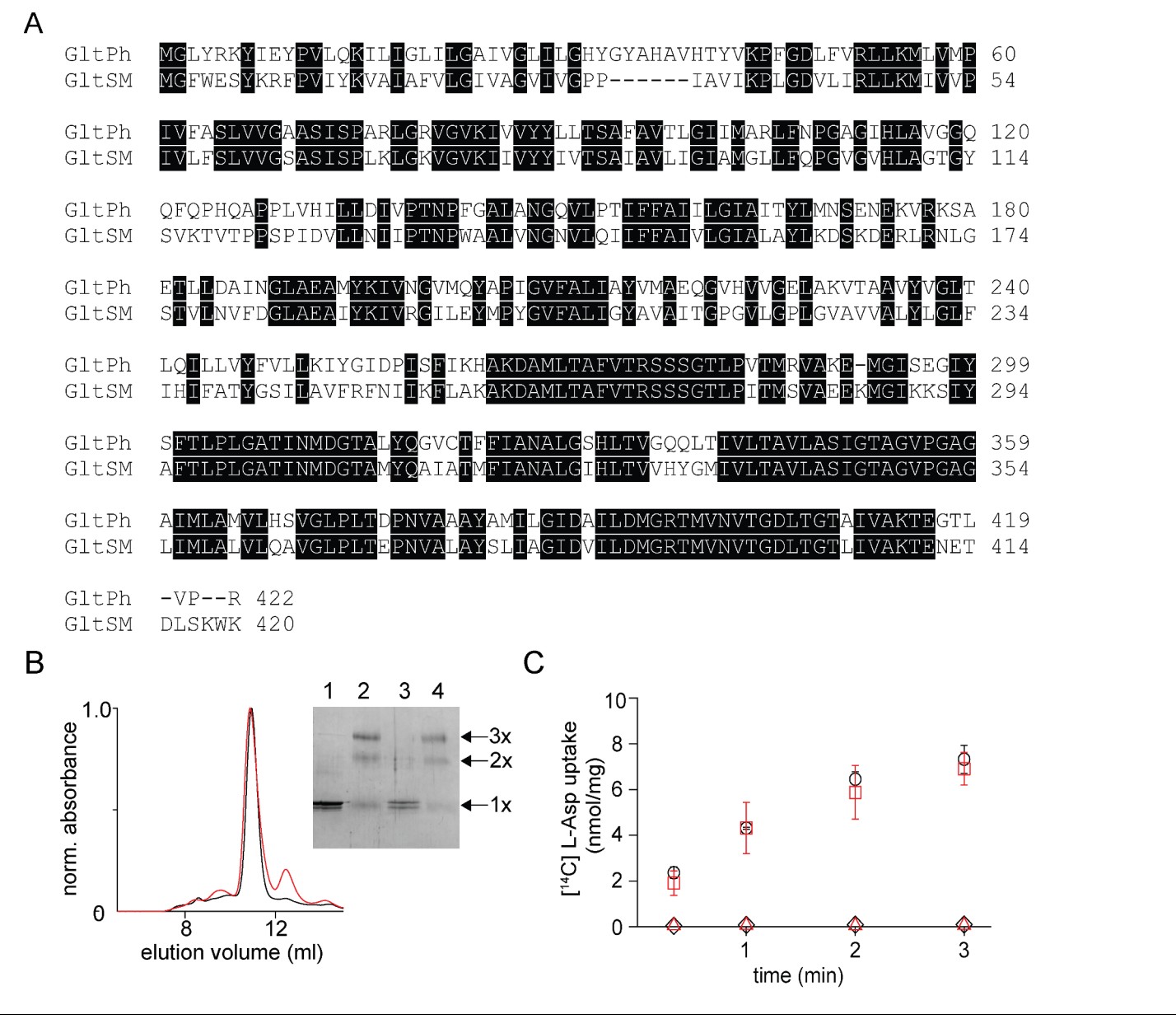

**Figure 5.** In vitro assembly of $Glt_{Sm}$. (**A**) Sequence alignment of $Glt_{Sm}$ with $Glt_{Ph}$ with identical residues highlighted. (**B**) Size exclusion chromatography of the native (black) and the reassociated $Glt_{Sm}$ (r-$Glt_{Sm}$, red). Inset, SDS-PAGE gel shows the glutaraldehyde cross-linking of the native and r-$Glt_{Sm}$. Native $Glt_{Sm}$ without (lane 1) and with glutaraldehyde crosslinking (lane 2) and r-$Glt_{Sm}$ without (Lane 3) and with glutaraldehyde crosslinking (Lane 4) are shown. The oligomeric nature of the protein band (1x, 2x, and 3x) is indicated. (**C**) Aspartate uptake assay. The time course of [$^{14}$C]-Asp uptake by native $Glt_{Sm}$ (black circles) and r-$Glt_{Sm}$ (red squares) in the presence of a $Na^+$ gradient. No uptake is observed for native $Glt_{Sm}$ (black diamonds) and r-$Glt_{Sm}$ (red triangles) in the absence of a $Na^+$ gradient. Error bars indicate SEM for $n \geq 3$.

DOI: https://doi.org/10.7554/eLife.36478.009

We further tested the procedure on the VcINDY and CLC-ec1 transporters. VcINDY is a homo-dimeric $Na^+$- coupled succinate transporter (**Figure 6A**)(**Mancusso et al., 2012**). Each subunit of VcINDY consists of 11 transmembrane helices that are arranged in two domains, a scaffold dimerization domain and a peripheral transport domain. The transport mechanism in VcINDY involves an elevator-like movement of the transport domain with respect to the scaffold domain (**Mulligan et al., 2016**). While the elevator mechanism of VcINDY is reminiscent of the mechanism in $Glt_{Ph}$, the topology and fold of the VcINDY transporter is very distinct (**Drew and Boudker, 2016**). The VcINDY

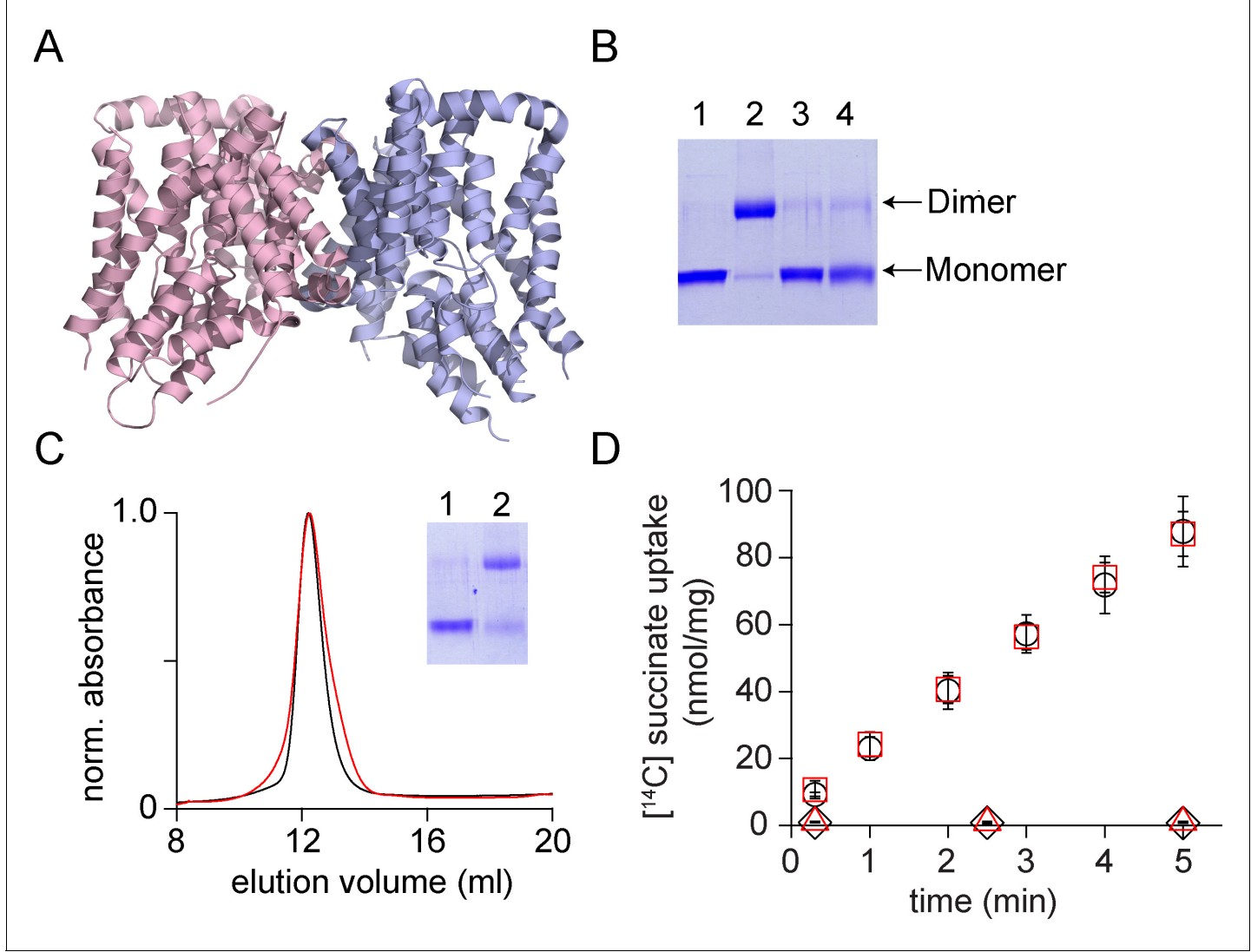

**Figure 6.** In vitro assembly of VcINDY. (**A**) Structure of VcINDY (pdb: 4f35) is shown in ribbon representation. (**B**) Dissociation of the VcINDY dimer by SDS. SDS-PAGE gel showing the native VcINDY without (lane 1), with glutaraldehyde crosslinking (lane 2) and the SDS dissociated VcINDY without (lane 3) and with glutaraldehyde crosslinking (lane 4). (**C**) Size exclusion chromatography of the native (black) and the reassociated VcINDY (r-VcINDY, red). Inset, SDS-PAGE gel showing the glutaraldehyde cross-linking of the peak fraction of r-VcINDY without (lane 1) and with glutaraldehyde crosslinking (lane 2). (**D**) Succinate uptake assay. The time course of [$^{14}$C]-succinate uptake by native VcINDY (black circles) and r-VcINDY (red squares) in the presence of a Na$^+$ gradient. No uptake is observed for native VcINDY (black diamonds) and r-VcINDY (red triangles) in the absence of a Na$^+$ gradient. Error bars indicate SEM for n $\geq$ 3.

DOI: https://doi.org/10.7554/eLife.36478.010

transporter is dimeric in the native state but migrates as a monomer on SDS-PAGE, indicating that SDS dissociates the transporter into monomers (*Figure 6B*). We used glutaraldehyde crosslinking to confirm complete dissociation of the VcINDY dimer by SDS. Crosslinked native VcINDY runs as a dimer on SDS-PAGE while VcINDY crosslinked after dissociation by SDS runs as a monomer. As anticipated, crosslinking following reassembly in lipid vesicles yielded a dimeric species. We solubilized the reassembled (r-)VcINDY protein in DDM detergent and purified it using a His-tag. Analysis by SEC showed that r-VcINDY had a similar elution profile to the native transporter (*Figure 6C*). The r-VcINDY transporter reconstituted into lipid vesicles mediated succinate uptake in the presence of a Na$^+$ ion gradient with a rate similar to the native transporter (*Figure 6D*).

CLC-ec1 is a Cl$^-$/H$^+$ antiporter; it is a homodimeric protein in which each subunit consists of 18 helical segments (*Figure 7A*)(*Dutzler et al., 2002*; *Accardi and Miller, 2004*). The CLC-ec1 fold has

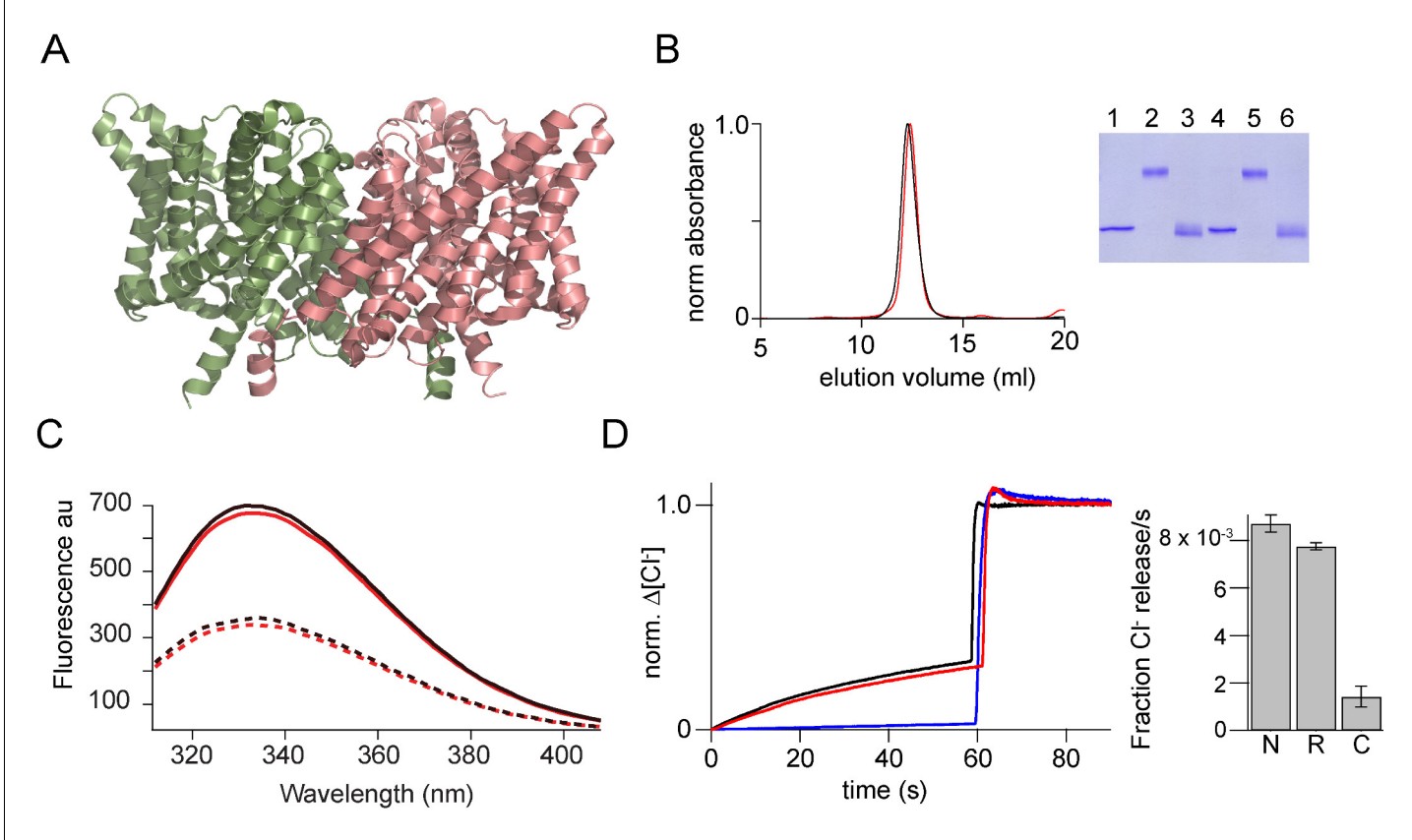

**Figure 7.** In vitro assembly of CLC-ec1. (**A**) Structure of CLC-ec1 (pdb: 1kpl) shown in ribbon representation. (**B**) Size exclusion chromatography of the native (black) and the in vitro assembled CLC-ec1 (r-CLC-ec1, red). Inset, SDS-PAGE gel showing the glutaraldehyde cross-linking of the peak fraction of native and r-CLC-ec1. Native CLC-ec1 without glutaraldehyde crosslinking (lane 1), with glutaraldehyde crosslinking before (lane 2) and after (Lane 3) treatment with 1% SDS. r-CLC-ec1 without glutaraldehyde crosslinking (lane 4), with glutaraldehyde crosslinking before (lane 5) and after (Lane 6) treatment with 1% SDS. (**C**) Fluorescence spectra of the native and r-CLC-ec1. Intrinsic fluorescence spectra (excitation at 295 nm) for the native (solid black) and r-CLC-ec1 (solid red) in 50 mMTris-HCl, pH 7.5, 200 mM NaCl, 0.25% DM. Dashed lines show the fluorescence spectra for the native (dashed black) and r-CLC-ec1 (dashed red) after the addition of 1% SDS. (**D**) Chloride transport assays. Left: Chloride efflux from liposomes containing native CLC-ec1 (Black), r-CLC-ec1 (red), and no protein control (blue). Bulk efflux was initiated at time zero by the addition of valinomycin. Triton X-100 was added after 60 s to release all of the intra-vesicular chloride. Chloride efflux was normalized to the total chloride concentration upon addition of Triton X-100. Right: Bar graph showing the normalized rate of chloride movement through liposomes containing native CLC-ec1 (N), r-CLC-ec1 (R), and no protein control (C). Normalized rates were calculated from the initial slope after addition of valinomycin from graphs such as those shown on the left. Error bars represent the SEM from six independent flux assays.

DOI: https://doi.org/10.7554/eLife.36478.011

an internal antiparallel structural repeat with the N-terminal nine helical segments structurally related to the C-terminal segments. The CLC-ec1 fold is very complex and thereby provides an excellent test for the general applicability of the in vitro reassembly protocol. Studies on the dimerization of CLC-ec1 have shown that it is one of the strongest membrane protein complexes known (*Chadda et al., 2016*). Thus, we used a more stringent protocol to ensure complete dissociation of the dimers. Specifically, the protein was precipitated with trichloroacetic acid and acetone, dissolved in trifluoroethanol: buffer A [$H_2O$ and 0.1% (v/v) trifluoroacetic acid] and lyophilized (*Devaraneni et al., 2011*). When the lyophilized protein was solubilized in 1% SDS, it migrated as a homogeneous monomer on SDS PAGE.

To reassemble the dimer, we diluted the SDS solubilized CLC-ec1 subunits into lipid vesicles. Following incorporation into lipid vesicles, glutaraldehyde cross-linking indicated the formation of dimers, suggesting the successful reassembly of CLC-ec1. The reassembled (r-) CLC-ec1 was solubilized in decyl-β-D-maltopyranoside (DM) detergent and purified from lipid vesicles using His$_6$ tag. SEC of r-CLC-ec1 produced an elution profile that was similar to the native CLC-ec1 (*Figure 7B*).

The intrinsic fluorescence spectra of the native CLC-ec1 shows an emission maxima at 333 nm (*Figure 7C*). Addition of SDS results in a 46% decrease in fluorescence intensity. The fluorescence spectra of the r-CLC-ec1 was similar to the native protein and showed a similar decrease in fluorescence intensity on the addition of SDS. The r-CLC-ec1 transporter reconstituted into lipid-vesicles-mediated chloride flux activity similar to the native protein (*Figure 7D*) suggesting that a fully functional transporter has been reassembled.

## Discussion

Here, we present a facile approach for the in vitro reassembly of heteromeric variants of homomeric membrane proteins. The method involves the assembly of the multimeric proteins from dissociated subunits in the context of lipid vesicles. Heteromeric proteins are obtained simply by mixing wild-type and mutant subunits. We established the approach using the Glt$_{Ph}$ transporter and then demonstrated its applicability to a number of other systems including Glt$_{Sm}$, VcINDY and CLC-ec1 transporters. Using biochemical, functional and single molecule approaches, we show that the reassembled transporters are indistinguishable from native proteins.

Heteromeric Glt$_{Ph}$ transporters allowed us to probe the communication between the Asp-binding sites. In typical binding experiments, the presence or absence of coupling between multiple binding sites in multimeric transporters is deduced from the shape of the binding isotherms and corresponding Hill coefficients (*Weiss, 1997*). The Hill coefficients that are greater or less than unity suggest, respectively, positive or negative cooperativity between the binding sites. However, Hill coefficients are relatively poorly determined and can be affected by baseline drifts and noise in the data. Furthermore, the existence of multiple coupled substrate binding sites within single subunits, as for example in BetP (*Ge et al., 2011*), may obscure the extent of inter-subunit coupling. The approach that we develop here allows us to circumvent these complications by assembling the reporter subunit together with test subunits that either can or cannot bind substrate with relevant affinity. Using this approach, we show that binding to the adjacent test subunits does not affect Asp binding to the reporter subunit thereby establishing unambiguously that the substrate-binding sites in Glt$_{Ph}$ are uncoupled.

Whether independent binding sites are a general feature of the multimeric transporters is not fully known, and our approach can provide a facile means of probing for such long-distance interactions between subunits. Interestingly, some transporters are assembled from multiple subunits with substrate-binding sites located at the interfaces, for example, the EmrE and the semi-SWEET transporters (*Chen et al., 2007*; *Latorraca et al., 2017*). The ability to break the intrinsic symmetry of these systems by assembling them from distinct subunits may open novel approaches to probe their mechanisms.

A multitude of spectroscopic approaches to probe the dynamics of transporters and channels has been developed in recent years. Most of them rely on site-specific labeling of introduced cysteine residues. DEER spectroscopy and smFRET microscopy specifically have been applied to study conformational flexibility and dynamics of diverse systems (*Joo et al., 2008*; *McHaourab et al., 2011*). A major challenge in the experimental design is identification of informative labeling sites. Because these approaches are based on distance measurements, an ideal labeling strategy would incorporate two labels at sites that show conformation-dependent distance changes. However, in multimeric proteins this effort is compounded by the presence of multiple subunits, each of which would carry cysteine substitutions. Thus, many of the successful experiments relied on measurements of inter-subunit distance changes (*Zou et al., 2009*; *Georgieva et al., 2013*; *Hänelt et al., 2013*; *Dastvan et al., 2016*; *Khantwal et al., 2016*). Clearly, these approaches have severe limitations when the structural transitions of interest occur within individual subunits. The ability to assemble a multimeric system of interest with single reporter subunits bearing cysteine mutations would significantly expand the repertoire of useful experiments and also facilitate and simplify analysis. We demonstrate the advantage of the reassembled heteromeric proteins by using DEER spectroscopy to probe the conformational state of Glt$_{Ph}$ transporter. Indeed, the distance distribution that was obtained for heteromeric Glt$_{Ph}$ bearing a single labeled subunit is dramatically simplified compared to a homomeric labeled protein, and allows facile determination of the conformational state of the labeled subunit. Similarly, the heteromeric transporters will be useful in introducing fluorescent probes at appropriate sites for smFRET studies of multimeric membrane transporters and channels.

For example, we expect significant simplification and increased resolution compared to approaches previously employed to study dynamics of Glt$_{Ph}$ (*Akyuz et al., 2013*; *Erkens et al., 2013*).

Many ion channels are homomeric in nature. Ion channels transition between different conformational states during function. Determining the structural nature of these conformational changes and how these changes are coupled among the individual ion channel subunits is important for understanding their mechanism. The approaches developed herein for the study of transporters can similarly be applied to homomeric ion channels to generate heteromeric variants. These heteromeric variants will provide the ability to alter the amino acid sequence of one or several of the subunits for functional or spectroscopic investigations and will potentially be very useful in probing the mechanism of action.

In our approach, SDS is used to dissociate the multimeric protein. SDS is not a strong denaturant for most membrane proteins and therefore we expect that the SDS-dissociated subunits are only partially unfolded (*Harris and Booth, 2012*). We also used a stringent dissociation process involving treatment with strong acids and organic solvents. Following this stringent dissociation protocol, we were able to observe successful reassembly for only two of the proteins in our test set, Glt$_{Ph}$ and CLC-ec1 while VcINDY and Glt$_{Sm}$ could not be reassembled. This observation suggests that the success of the disassembly/reassembly depends on the gentle dissociation of the multimeric proteins without extensive unfolding of the subunits. Further, lipid bilayers were required for the reassociation process, and a mere dilution of the SDS in the presence of mild detergents did not result in successful reassembly. We speculate that incorporation of the dissociated subunits into lipid bilayers concentrates and orients the subunits to facilitate the assembly.

The overall mechanism of coupled refolding and assembly that takes place in the lipid bilayers is not clear. For CLC-ec1, it was shown that monomers are stable (*Robertson et al., 2010*), and thus, it is possible that these transporters first assume a native-like fold and then assemble into dimers. The trimeric Glt$_{Ph}$ forms a bowl-like structure in the outward facing state (*Yernool et al., 2004*). The bowl extends approximately half way across the bilayer; it is lined with polar residues and is most certainly filled with water. It seems unlikely that the conformation of dissociated Glt$_{Ph}$ monomers is entirely native-like because it would lead to the exposure of the polar regions of the bowl to the hydrophobic milieu of the membrane. Thus, it seems likely that reaching the native conformation is coupled with the trimer assembly in this protein. Our reassembly protocols will be useful to probe the mechanisms of multimeric membrane protein assembly. For example, these processes have been shown to have specific lipid requirements (*Gupta et al., 2017*). The ability to assemble multimeric membrane proteins in defined lipid environments will facilitate an investigation of how lipid molecules participate in the multimerization process.

While our experiments have only focused on bacterial and archaeal transporters, it will be of great interest to determine whether this simple protocol is applicable also to eukaryotic proteins. Using this procedure for multimeric eukaryotic proteins will involve identifying appropriate detergents for mild dissociation and appropriate compositions of lipid bilayers for reassociation.

## Materials and methods

**Key resources table**

| Reagent type (species) or resource | Designation | Source or reference | Identifiers | Additional information |
|---|---|---|---|---|
| Gene (*Pyrococcus horikoshii*) | GltPh | 10.1038/nature03018 | Uniprot ID: O59010 | |
| Gene (*Vibrio cholerae*) | VcINDY | 10.1038/nature11542 | Uniprot ID: Q9KNE0 | |
| Gene (*Escherichia coli*) | CLC-ec1 | 10.1085/jgp.200308935 | Uniprot ID: P37019 | |
| Gene (*Staphylothermus marinus*) | GltSm | 10.1111/febs.12105 | Uniprot ID: A3DPQ3 | |
| Recombinant DNA reagent | pBCH/G4-GltPh | 10.1038/nature03018 | | |

*Continued on next page*

*Continued*

| Reagent type (species) or resource | Designation | Source or reference | Identifiers | Additional information |
|---|---|---|---|---|
| Recombinant DNA reagent | pET-VcINDY | 10.1038/nature11542 | | |
| Recombinant DNA reagent | pASK-CLC-ec1 | 10.1085/jgp.200308935 | | |
| Recombinant DNA reagent | pBAD-GltSm | This study | | GltSm from the GltSm-GFP fusion gene (ref: 10.1111/febs.12105) was cloned into a pBAD-HisA vector (Fisher Scientific) |
| Chemical compound, drug | Asolectin | Avanti Polar Lipids | Cat # 541601G | |
| Software, algorithm | Pymol | PyMOL Molecular Graphics System, Schrödinger, LLC | RRID:SCR_000305 | www.pymol.org |
| Software, algorithm | Matlab | Mathworks | RRID:SCR_001622 | www.mathworks.com |
| Software, algorithm | Origin | Originlab | RRID:SCR_014212 | www.originlab.com |

## Glt$_{Ph}$

### Native expression

Glt$_{Ph}$ constructs used in this study were expressed from the pBCH/G4 vector (kindly provided by Dr. Eric Gouaux) in *Escherichia coli* TOP10 cells (Fisher Scientific)(*Yernool et al., 2004*). Protein expression and membrane preparation was carried out as described (*Focke et al., 2015*). The membrane vesicles were solubilized using dodecyl-β-D-maltopyranoside [DDM, 2% (w/v)] and Glt$_{Ph}$ was purified by metal affinity chromatography (Ni NTA resin, Qiagen) and size exclusion chromatography (SEC) as described (*Focke et al., 2015*). SEC was carried out on a Superdex S-200 column (GE Biosciences) using 20 mM HEPES-NaOH pH 7.5, 200 mM NaCl, and 0.1% (w/v) DDM as the column buffer.

### Dissociation and reassociation of Glt$_{Ph}$

Dissociation of Glt$_{Ph}$ into subunits was carried out by the addition of 1% (w/v) SDS, 0.1 M DTT, 1 mM EDTA and incubation at 45°C for 1 hr. Reassociation was carried out by a 10-fold dilution of the SDS-dissociated subunits into 20 mg/ml Asolectin vesicles in lipid buffer (20 mM HEPES-NaOH pH 7.5, 200 mM NaCl, 10 mM DTT, 1 mM Asp) and incubated at room temperature for 5 hr. For purification of the reassociated Glt$_{Ph}$ (r- Glt$_{Ph}$), the lipid vesicles were dialyzed against 20 mM HEPES-NaOH pH 7.5, 200 mM NaCl for the removal of DTT and EDTA. r-Glt$_{Ph}$ was purified from the lipid vesicles as described for the native protein. For the detergent control, the SDS dissociated Glt$_{Ph}$ was diluted 10-fold into 2% (w/v) DDM in lipid buffer. Following incubation at room temperature for 5 hr, the solution was dialyzed against 20 mM HEPES-NaOH pH 7.5, 200 mM NaCl, 0.1% (w/v) DDM and the protein was purified as described.

### Glutaraldehyde crosslinking

The oligomeric state of the proteins was assessed using chemical crosslinking with 0.1% (w/v) glutaraldehyde for 15 min at room temperature. Crosslinking reaction was quenched by the addition of 100 mM Tris. The samples were electrophoresed on a 12% SDS-PAGE gel and the proteins were visualized by staining with Coomassie Blue.

### Aspartate-binding assays

Aspartate-binding assays were carried out on Glt$_{Ph}$ with a L130W substitution (*Boudker et al., 2007*). Prior to the binding assay, the Glt$_{Ph}$ sample was extensively dialyzed against the assay buffer [20 mM Tris-HEPES pH 7.4, 200 mM Choline Chloride, 0.1% (w/v) DDM] to remove any bound Asp. The binding assays were carried out using ~100 nM of Glt$_{Ph}$ at 30°C in assay buffer containing 10 mM NaCl. Binding of Asp was monitored by the change in the fluorescence emission at 334 nm

following excitation at 295 nm. The fraction bound ($F_b$) was calculated by normalizing the change in fluorescence following Asp addition to the maximum change observed and $K_D$ was determined by a fit to the equation: $F_b = \frac{[Asp]}{K_D+[Asp]}$

## Aspartate transport assays

For Asp transport assays, the native and r-Glt$_{Ph}$ was reconstituted at into lipid vesicles comprised of a 3: 1 ratio of *Escherichia coli* Polar lipids to 1-palmitoyl-2-oleoyl-glycero-3phosphatidylcholine (POPC) at 6 µg of protein/mg of lipid as previously described (*Ryan et al., 2009*). The transport assays were carried out in 20 mM Tris-HEPES pH 7.5, 200 mM NaCl, 1 µM valinomycin and 100 nM $^{14}$C Asp (Moravek Biochemicals) at room temperature as previously described (*Focke et al., 2015*). Background levels of Asp transport was determined in the absence of NaCl, with 100 mM KCl in the assay buffer.

## Single molecule experiments

Native and r- Glt$_{Ph}$ used for the smFRET experiments carried the N378C substitution. The native and r-Glt$_{Ph}$ molecules were exchanged into Buffer A [20 mM Hepes/NaOH, pH 7.4, 200 mM NaCl, 0.1 mM L-aspartate, 0.1 mM Tris(2-carboxyethyl)phosphine and 1 mM DDM] using SEC and the proteins were labeled at a concentration of 20 µM in Buffer A with a mixture of maleimide-activated Cy3, Cy5 and biotin-PEG$_{11}$ at 50, 100 and 20 µM final concentrations, respectively (molar ratio 1:2:0.4), as before (*Akyuz et al., 2013*). Single-molecule experiments were performed using a home-built, prism-based total internal reflection fluorescence instrument constructed around a Nikon TE2000 Eclipse inverted microscope body. Individual molecules were surface immobilized within a streptavi-din-coated, passivated microfluidic via the biotin-PEG$_{11}$ moiety (*Munro et al., 2007*; *Akyuz et al., 2013*). All imaging experiments were performed in a buffer containing: 200 mM NaCl and 0.1 mM aspartate, 20 mM Tris-HEPES pH 7.4, 1 mM DDM, 5 mM β-mercaptoethanol, 1 mM cyclooctate-traene, an enzymatic oxygen scavenger system comprising 1 unit/ml glucose oxidase (Sigma), 8 units/ml catalase (Sigma) and 0.1% glucose (*Dave et al., 2009*).

Acquired Cy5 intensities ($I_{Cy5}$) from native and r-Glt$_{Ph}$ molecules were analyzed in MATLAB (Math-works) using the SPARTAN software package (*Juette et al., 2016*) (available at http://www.scottc-blanchardlab.com/software) and plotted in Origin (OriginLab). FRET trajectories obtained were calculated from the acquired intensities, $I_{Cy3}$ and $I_{Cy5}$, using the formula FRET = $I_{Cy5}/(I_{Cy3} + I_{Cy5})$. Tra-jectories for further analysis were selected within the SPARTAN software environment according to the following criteria: a single catastrophic photobleaching event; over 8:1 signal-to-background noise ratio; a FRET lifetime of at least 5 s. Population contour plots were constructed by superimpos-ing the FRET data from individual traces. Histograms of these population data were fit to Gaussian functions in Origin (OriginLab).

## Assembly of heteromeric Glt$_{Ph}$

The wild-type Glt$_{Ph}$ construct or with the R397A substitution were expressed and purified as described. The His$_6$ tag present was removed by proteolysis with Thrombin overnight at room tem-perature. The complete removal of the His$_6$ tag was confirmed by SDS-PAGE and the protein was purified by SEC as described. Glt$_{Ph}$ protein with the L130W substitution was similarly expressed and purified but in this case the His$_6$ tag was not removed.

Prior to the mixing experiment, the concentration of the proteins was determined by measuring the absorbance at 280 nm. Since accurate determination of the protein concentration is important, the protein concentrations were confirmed by running a serial dilution of the proteins on a SDS-PAGE gel and comparing the intensity of the protein bands after Coomassie Blue staining. The wild type and the R397A Glt$_{Ph}$ proteins were mixed with the L130W Glt$_{Ph}$ protein in a 10: 1 ratio. The protein was then dissociated and reassociated as described. The presence of a His$_6$ tag on the L130W subunit allows the purification of the Glt$_{Ph}$ heterotrimers with a L130W subunit. The Glt$_{Ph}$ heterotrimers were further purified by SEC. Asp-binding assays for the heterotrimeric Glt$_{Ph}$ were car-ried out as described.

## DEER experiments

The Glt$_{Ph}$ construct with cysteine substitutions at 216 and 294 positions was expressed and purified as described above. Following purification, the protein in detergent solution was labelled with the MTSL spin label (1-oxyl-2,2,5,5-tetramethylpyrrolidin-3-yl) methyl methanethiosulfonate; Toronto Research Chemicals) at 20:1 spin label to channel molar ratio. Following overnight labeling at 4°C, the excess label was removed using SEC as described (*Focke et al., 2011*). Heteromeric Glt$_{Ph}$ transporters, in which only a single subunit carried the 216, 294 Cys substitutions, were assembled as previously described and spin labeled.

Prior to DEER measurements, the buffer was exchanged to 200 mM NaCl or 200 mM NaCl/300 µM aspartate, 2 mM DDM, 20 mM HEPES pH 7.4, and ca. 80–85% D$_2$O. (Stock solutions of buffer components in H$_2$O were mixed and diluted with D$_2$O to the final concentrations). Subsequently, ca. 96 µM GltPh monomer solution (i.e. about 32 µM trimer) was mixed with glycerol-d$_8$ to the final glycerol concentration of 20% (w/v). Sample volumes of about 20 µl were loaded into custom-sized glass capillary tubes (o.d. ~2.6 mm, Wilmad LabGlass, Inc.) and plunge-frozen in liquid N$_2$ for DEER measurements. All measurements were performed at 60 K using a home-built Ku-band pulse EPR spectrometer (*Borbat et al., 1997*) operating at 17.3 GHz as described previously (*Georgieva et al., 2013*; *Georgieva et al., 2015*). The standard four-pulse DEER experiment (*Jeschke, 2002*) used for detection π/2-$t_1$-π-$t_2$-π pulse sequence with respective pulse widths of 16 ns, 32 ns and 32 ns. A 32 ns pump π-pulse, applied at the center peak of nitroxide ESR spectrum, was used throughout all measurements. The frequency separation between detection and pump pulses was 70 MHz to position the detection pulses at the low-field edge of the nitroxide spectrum. Homogeneous (log-linear) signal background decay was removed from the raw DEER data and the background-subtracted DEER signals were normalized to read modulation depths at zero evolution time as described previously. (*Georgieva et al., 2015*) Interspin distances were reconstructed from the background-subtracted DEER data using the L-curve Tikhonov regularization method (*Chiang et al., 2005a*) and refined by the maximum entropy method (*Chiang et al., 2005b*).

## Glt$_{Sm}$

Glt$_{Sm}$ was amplified by PCR from a plasmid carrying the gene for a Glt$_{Sm}$-GFP fusion protein (*Jaehme and Michel, 2013*) and cloned into a pBAD-HisA vector (Fisher Scientific) with a N-terminal His$_6$ tag. The Glt$_{Sm}$ vector was transformed into TOP10 cells (Fisher Scientific) for protein expression. Protein expression and purification of the Glt$_{Sm}$ protein was carried out as described for Glt$_{Ph}$. Dissociation and reassociation of Glt$_{Sm}$ was carried out using the same protocol used for Glt$_{Ph}$. The native and reassociated Glt$_{Sm}$ was reconstituted into lipid vesicles comprised of a 3: 1 ratio of *Escherichia coli* polar lipids to POPC at 3 µg of protein/mg of lipid and transport assays were carried out in 20 mM Tris-HEPES pH 7.5, 200 mM NaCl, 1 µM valinomycin and 100 nM $^{14}$C Asp at 30°C as previously described for the Glt$_{Ph}$ transporter.

## VcINDY

### Native expression

VcINDY was expressed from a pET vector (kindly provided by Dr. Joseph Mindell) in *Escherichia coli* BL21-AI (Fisher Scientific) cells as described (*Mancusso et al., 2012*; *Mulligan et al., 2014*). Following expression, cells were pelleted, suspended in 50 mM Tris-HCl pH 7.5, 200 mM NaCl, 0.25 M sucrose, 1 mM MgCl$_2$ and membranes were prepared as previously described (*Devaraneni et al., 2011*). The membranes were solubilized in 2% (w/v) DDM and the VcINDY protein was purified using metal affinity chromatography (Talon, Clontech) followed by SEC. SEC was carried out on a Superdex S200 column (GE Biosciences) using 50 mM HEPES-NaOH (pH 7.5), 200 mM NaCl, 1 mM DTT, 0.1 mM EDTA and 0.1% (w/v) DDM as the column buffer.

### Dissociation and reassociation of VcINDY

Dissociation of VcINDY into subunits was carried out by the addition of SDS to 1% (w/v), 0.1 M DTT, 1 mM EDTA and incubation at 45°C for 30 min. Reassociation was carried out by a 10-fold dilution of the SDS-dissociated subunits into lipid vesicles (20 mg/ml Asolectin in 50 mM HEPES-NaOH, 50 mM Citrate, 100 mM NaCl, 10 mM DTT) and incubated overnight at room temperature. For purification of the reassociated (r-) VcINDY, the lipid vesicles were dialyzed against 50 mM Tris-HCl, pH 7.5,

200 mM NaCl to remove the DTT. The lipid vesicles were solubilized and r-VcINDY was purified as described for the native protein.

## Succinate transport assays

The native and the r-VcINDY were reconstituted into liposomes consisting of a 3:1 ratio of *Escherichia coli* Polar lipids to POPC (*Mulligan et al., 2014*). The lipids obtained as chloroform solutions, were dried and re-suspended at a concentration of 10 mg/ml in 10 mM HEPES-KOH, pH 7.5, 100 mM KCl. The lipid solution was subjected to five freeze–thaw cycles with liquid $N_2$ and then extruded through a 400-nm filter. The lipid vesicles were partially solubilized by the addition of decyl-β-D-maltopyranoside [DM, 0.5% (w/v)]. Protein was added to the lipids at a ratio of 3 μg protein/mg lipid along with another aliquot of DM to bring the detergent concentration to 1.0% (w/v). The protein-lipid mixture was incubated at room temperature for 3–4 hr with gentle shaking and then the detergent was gradually removed, and proteoliposomes were formed by multiple additions of Biobeads (Bio-Rad Laboratories) over 24 hr. The proteoliposomes were separated from the Biobeads, collected by centrifugation, frozen in aliquots using liquid $N_2$ and stored at −80°C.

For transport assays, the previously frozen proteoliposomes were thawed and centrifuged at 175,000 g for 70 min at 4°C. The pelleted proteoliposomes were re-suspended in the low $Na^+$ buffer [20 mM Tris-HEPES (pH 7.5), 199 mM KCl, 1 mM NaCl] at 10 mg/ml lipid, subjected to two freeze-—thaw cycles with liquid $N_2$, and extruded through a 400 nm filter. The proteoliposomes were then centrifuged and re-suspended in the low $Na^+$ buffer at 100 mg/ml lipid. The uptake reaction was initiated by diluting the proteoliposomes 100-fold into the assay buffer [20 mM Tris-HEPES (pH 7.5), 100 mM KCl, 100 mM NaCl and 1 mM $^{14}C$-Succinate (Moravek Biochemicals)] at room temperature. For each time point, a 200 μL aliquot was removed and diluted 10-fold into ice-cold quench buffer [20 mM Tris-HEPES (pH 7.5), 100 mM choline chloride] followed by filtration over nitrocellulose filters (0.22 μm, Millipore). Filters were washed thrice with 2 ml of ice-cold quench buffer and assayed for radioactivity. Background levels of $^{14}C$-Succinate uptake were determined in the absence of a sodium gradient (1 mM $Na^+$ on both sides).

# CLC-ec1

## Dissociation of the CLC-ec1 dimer

CLC-ec1 containing a C-terminal polyhistidine tag was overexpressed in *Escherichia coli* BL21 (DE3) cells and purified as described (*Accardi et al., 2004*). Dissociation of the CLC-ec1 was carried out as described for the $K_vAP$ channel (*Devaraneni et al., 2011*). Briefly, Triton X-100 was added to the CLC-ec1 solution to 2% (v/v), and the protein was precipitated by the addition of trichloroacetic acid to 15% (w/v) and incubated at 4°C for 30 min. The protein precipitate was collected by centrifugation, washed twice with acetone and 0.1% TFA (trifluoroacetic acid), and then solubilized in 50% TFE (trifluoroethanol) and 0.1% TFA. The TFE solution was lyophilized to provide the dissociated CLC-ec1 that was used for in vitro assembly.

## In vitro assembly of CLC-ec1

The lyophilized CLC-ec1 was dissolved in 50 mM HEPES-NaOH pH 7.4, 200 mM NaCl, 10 mM EDTA, 0.1 M DTT, 1% (w/v) SDS and diluted 10-fold into lipid vesicles (20 mg/ml asolectin in 50 mM HEPES-NaOH, 200 mM NaCl, 10 mM DTT) and incubated overnight at room temperature. For purification, the lipid vesicles were initially dialyzed against 50 mM Tris-HCl, pH 7.5, 200 mM NaCl to remove the DTT and EDTA. The lipid vesicles were then solubilized and the in vitro assembled CLC-ec1 (r-CLC-ec1) was purified as described for the native protein.

## Chloride flux assays for CLC-ec1

The native and r-CLC-ec1 channels were reconstituted into liposomes and $Cl^-$ flux assays were carried out similarly to that previously described (*Walden et al., 2007*) with some minor differences. Briefly, ClC-ec1 was reconstituted into *E. coli* polar lipids at a concentration of 0.4 μg of protein per mg lipid. Detergent was removed to form liposomes by dialysis against 300 mM KCl, 25 mM citrate pH 4.5. For flux assays, vesicles were freeze-thawed five times in dry ice/acetone and extruded through a 400 nm filter. Extra-vesicular solution was exchanged by centrifuging 60 μl of extruded vesicles through a 1.5 ml Sephadex G-50 column equilibrated in Flux Buffer (100 mM $K_2SO_4$, 0.05

mM KCl, 25 mM anhydrous citrate pH 4.5 with NaOH). The vesicles were then diluted into Flux buffer and extra-vesicular chloride concentration was monitored with a Ag/AgCl electrode. Chloride efflux was initiated with ~5.8 μM valinomycin, and after an additional 1 min, vesicles were broken upon with addition of ~0.13% Triton X-100. Flux was normalized to the total amount of chloride release after addition of Triton X-100.

## Acknowledgements

This research was supported by the grants from the NIH: R01 GM087546 (FV), R37NS085318 (OB, SB and FV), P41GM103521 (JHF, ACERT) and R01 GM123779 (JHF and ERG). PJF was supported by a postdoctoral fellowship from the American Heart Association (12POST11910068).

## Additional information

### Competing interests

Olga Boudker: Reviewing editor, *eLife*. The other authors declare that no competing interests exist.

### Funding

| Funder | Grant reference number | Author |
| --- | --- | --- |
| National Institute of General Medical Sciences | R01 GM087546 | Francis I Valiyaveetil |
| Howard Hughes Medical Institute | | Olga Boudker |
| National Institute of Neurological Disorders and Stroke | R37 NS085318 | Scott C Blanchard Olga Boudker Francis I Valiyaveetil |
| National Institute of General Medical Sciences | P41GM103521 | Jack H Freed |
| American Heart Association | 12POST1910068 | Paul J Focke |
| National Institute of General Medical Sciences | R01 GM123779 | Elka R Georgieva Jack H Freed |

The funders had no role in study design, data collection and interpretation, or the decision to submit the work for publication.

### Author contributions

Erika A Riederer, Kimberly Matulef, Investigation, Writing—original draft; Paul J Focke, Nurunisa Akyuz, Investigation; Elka R Georgieva, Formal analysis, Investigation, Writing—original draft; Peter P Borbat, Investigation, Writing—original draft, Writing—review and editing; Jack H Freed, Supervision, Writing–review and editing; Scott C Blanchard, Formal analysis, Investigation, Writing—review and editing; Olga Boudker, Supervision, Investigation, Writing—original draft, Writing—review and editing; Francis I Valiyaveetil, Conceptualization, Funding acquisition, Investigation, Methodology, Writing—original draft, Project administration, Writing—review and editing

### Author ORCIDs

Erika A Riederer https://orcid.org/0000-0003-1011-6536
Kimberly Matulef http://orcid.org/0000-0002-5011-9064
Jack H Freed https://orcid.org/0000-0003-4288-2585
Scott C Blanchard http://orcid.org/0000-0003-2717-9365
Olga Boudker https://orcid.org/0000-0001-6965-0851
Francis I Valiyaveetil http://orcid.org/0000-0002-3387-8018

### Decision letter and Author response

Decision letter https://doi.org/10.7554/eLife.36478.014

Author response https://doi.org/10.7554/eLife.36478.015

## Additional files

### Supplementary files
• Transparent reporting form
DOI: https://doi.org/10.7554/eLife.36478.012

### Data availability
All data generated or analyzed during this study are included in the manuscript and supporting files.

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
