## [Decision Letter]

Thank you for submitting your article "A Facile Approach for the in vitro Assembly of Multimeric Membrane Transport Proteins" for consideration by *eLife*. Your article has been favorably evaluated by Richard Aldrich (Senior Editor) and three reviewers, one of whom, Kenton J Swartz (Reviewer #1), is a member of our Board of Reviewing Editors.

The reviewers have discussed the reviews with one another and the Reviewing Editor has drafted this decision to help you prepare a revised submission.

Summary:

The production of multimeric complexes of membrane proteins of defined subunit composition has been a challenge for investigating many fundamental aspects of membrane protein function, including stoichiometry of ligand binding and cooperative interactions between subunits, as well as structural dynamics using FRET or EPR. The conventional approach of concatenating subunits to achieve the desired mixture has often been problematic because concatemers do not always faithfully assemble. In this manuscript, the Valiyaveetil laboratory reports a method for producing functional multimeric transporters of defined subunit combinations by first partially denaturing with SDS to achieve single subunits, and then refolding to form multimers using lipid vesicles. Although this work is a natural extension of the group's successful semi-synthesis of the glutamate transporter Glt_Ph_, where they discovered that subunits could be folded efficiently when diluting detergents in the presence of vesicles, the demonstration that defined subunit mixtures can be achieve is a significant advance. They show that they can produce trimeric Glt_Ph_ that is competent to bind Asp with high affinity, can transport the amino acid in a Na^+^-dependent fashion, and that exhibits dynamic behavior in smFRET experiments that resembles what is observed with more traditional methods of producing and purifying Glt_Ph_. They then show that heteromeric complexes can be produced that contain a single subunit with a Trp mutant that reports on Asp binding, and that Asp binding to that subunit is unaffected by mutations in the other two subunits that effectively ablate Asp binding. Although other evidence supports the idea that the three subunits in Glt_Ph_ transport Asp independently, this is the most direct demonstration to date. They also show that they can label a single subunit with DEER probes and that they can acquire greatly simplified EPR spectra with a single relatively tight distance distribution. Finally, the authors demonstrate that the refolding/assembly approach also works for another transporter VcINDY, as well as for a CLC proton/Cl^-^ antiporter.

Essential revisions:

1) Although the work on the Glt_Ph_ transporter is extensive, aspects of the work on CLC-ec1 and VcINDY appear less rigorous. 'Notably, TCA treatments of CLC-ec1 likely leads to extensive if not complete unfolding […].these results also show that the CLC-ec1 transporter undergoes a coupled in vitro folding and dimerization [...]' This statement is not supported by the data presented, as the authors do not attempt to measure the unfolding of CLC-ec1 using spectroscopy, or measure the refolding. We think this is an area that many readers would find interesting, and pertains to the effectiveness of trying to refold membrane proteins using this approach. Is it possible for the authors to show what they mean by 'successful refolding' (Discussion). Is this a measure of monodispersity or function?

2) Although the manuscript is well-written, the authors should carefully go through the document to fix a few awkward sentences. Examples include the fourth paragraph of the Discussion where 'An' should be changed to 'The".

---

## [Author Response]

Essential revisions:1) Although the work on the Glt_Ph_ transporter is extensive, aspects of the work on CLC-ec1 and VcINDY appear less rigorous. 'Notably, TCA treatments of CLC-ec1 likely leads to extensive if not complete unfolding […] these results also. show that the CLC-ec1 transporter undergoes a coupled in vitro folding and dimerization [...]' This statement is not supported by the data presented, as the authors do not attempt to measure the unfolding of CLC-ec1 using spectroscopy, or measure the refolding. We think this is an area that many readers would find interesting, and pertains to the effectiveness of trying to refold membrane proteins using this approach. Is it possible for the authors to show what they mean by 'successful refolding' (Discussion). Is this a measure of monodispersity or function?

We have demonstrated by chemical cross-linking that the native CLC-ec1 is a dimer and is dissociated to a monomeric state by treatment with SDS (Figure 7B). We show by size exclusion chromatography and cross-linking that the re-associated (r-) CLC-ec1 is dimeric similar to the native transporter (Figure 7B). We have now included new data showing that the fluorescence properties of CLC-ec1 following treatment with SDS is distinct from the native protein while the r-CLC-ec1 has identical fluorescence properties to the native protein (Figure 7C). Further, we have demonstrated that the r-CLC-ec1 has similar functional properties to native CLC-ec1 (Figure 7D). These data therefore establish that the re-assembly process using lipid vesicles results in the conversion of CLC-ec1 from a monomeric to a natively folded dimeric state.

It is quite likely that SDS treatment in addition to dissociating the CLC-ec1 dimer into monomeric subunits also causes a partial unfolding of the subunits. Further, our dissociation procedure for CLC-ec1 also included a TCA/organic solvent precipitation and lyophilization step prior to dissolution in SDS. We therefore inferred that the CLC-ec1 subunits following this stringent dissociation procedure are extensively unfolded.

We have not determined the extent of unfolding of CLC-ec1 following this procedure. It is very challenging to determine the extent of unfolding for an α-helical membrane protein such as CLC-ec1. The challenge is that additives such as SDS or trifluoroethanol, which are required for maintaining the unfolded/dissociated protein in solution, favor the formation of an α-helical state. As we do not know the extent of unfolding, we have deleted the following lines from the manuscript:

“Notably, TCA treatment of CLC-ec1 likely leads to extensive if not complete unfolding of the transporter. Thus, these results also show that the CLC-ec1 transporter undergoes a coupled in vitro folding and dimerization from an extensively unfolded state.”

We have also added a new panel (panel C) to Figure 7 showing that the fluorescence properties of the native and r-CLC-ec1 are similar to each other and different from the fluorescence properties of the proteins after treatment with SDS.

While we are presently not certain whether the procedure results in refolding of CLC-ec1 from an extensively unfolded state, our data clearly establish that the procedure does achieve the stated goal of in vitro assembly of a natively folded CLC-ec1 dimer from a dissociated state.

2) Although the manuscript is well-written, the authors should carefully go through the document to fix a few awkward sentences. Examples include the fourth paragraph of the Discussion where 'An' should be changed to 'The".

We thank the reviewers for pointing out a few instances. We have re-read the manuscript and made the changes to improve readability.

A minor change in the manuscript is that the Supplementary Figure 4 in the original submission is now included in the manuscript as Figure 5. This change was required due to the way that *eLife* handles figure supplements.